# Ex Vivo Anti-Leukemic Effect of Exosome-like Grapefruit-Derived Nanovesicles from Organic Farming—The Potential Role of Ascorbic Acid

**DOI:** 10.3390/ijms242115663

**Published:** 2023-10-27

**Authors:** Germana Castelli, Mariantonia Logozzi, Davide Mizzoni, Rossella Di Raimo, Annamaria Cerio, Vincenza Dolo, Luca Pasquini, Maria Screnci, Tiziana Ottone, Ugo Testa, Stefano Fais, Elvira Pelosi

**Affiliations:** 1Department of Oncology and Molecular Medicine, Istituto Superiore di Sanità, 00161 Rome, Italy; germana.castelli@iss.it (G.C.); mariantonia.logozzi@iss.it (M.L.); davide@exolabitalia.com (D.M.); rossella@exolabitalia.com (R.D.R.); annamaria.cerio@iss.it (A.C.); ugo.testa@iss.it (U.T.); 2ExoLab Italia, Tecnopolo d’Abruzzo, 67100 L’Aquila, Italy; 3Department of Clinical Medicine, Public Health, Life and Environmental Sciences, University of L’Aquila, 67100 L’Aquila, Italy; vincenza.dolo@univaq.it; 4Core Facilities, Istituto Superiore di Sanità, 00161 Rome, Italy; luca.pasquini@iss.it; 5Banca Regionale Sangue Cordone Ombelicale, UOC Immunoematologia e Medicina Trasfusionale, Policlinico Umberto I, 00161 Rome, Italy; m.screnci@policlinicoumberto1.it; 6Department of Biomedicine and Prevention, Tor Vergata University, 00133 Rome, Italy; tiziana.ottone@uniroma2.it; 7Santa Lucia Foundation, IRCCS, Neuro-Oncohematology, 00179 Rome, Italy

**Keywords:** grapefruit nanovesicles, ascorbic acid (vitamin C), acute myeloid leukemia (AML), oxidative stress

## Abstract

Citrus fruits are a natural source of ascorbic acid, and exosome-like nanovesicles obtained from these fruits contain measurable levels of ascorbic acid. We tested the ability of grapefruit-derived extracellular vesicles (EVs) to inhibit the growth of human leukemic cells and leukemic patient-derived bone marrow blasts. Transmission electron microscopy and nanoparticle tracking analysis (NTA) showed that the obtained EVs were homogeneous exosomes, defined as exosome-like plant-derived nanovesicles (ELPDNVs). The analysis of their content has shown measurable amounts of several molecules with potent antioxidant activity. ELPDNVs showed a time-dependent antiproliferative effect in both U937 and K562 leukemic cell lines, comparable with the effect of high-dosage ascorbic acid (2 mM). This result was confirmed by a clear decrease in the number of AML blasts induced by ELPDNVs, which did not affect the number of normal cells. ELPDNVs increased the ROS levels in both AML blast cells and U937 without affecting ROS storage in normal cells, and this effect was comparable to ascorbic acid (2 mM). With our study, we propose ELPDNVs from grapefruits as a combination/supporting therapy for human leukemias with the aim to improve the effectiveness of the current therapies.

## 1. Introduction

Acute myeloid leukemia (AML) is a heterogeneous clonal hematopoietic stem cell malignancy characterized by the uncontrolled proliferation and accumulation of immature progenitors in the bone marrow and peripheral blood, with arrested differentiation leading to suppression of normal hematopoiesis [1,2]. Consistent progress in DNA sequence technology has provided a dramatic improvement in the molecular characterization of these leukemias, showing their heterogeneity. AML is heterogeneous for its morphology, immunophenotype, molecular and cytogenetic abnormalities, epigenetic signatures and patient outcomes [1,2]. Current treatment for AML patients involves a combination of chemotherapy, the use of hypomethylating agents and/or hematopoietic stem cell transplantation [1,2]. More recently, targeted therapies have been introduced for the treatment of some AML subtypes [1,2]. The survival of AML patients is related to clinical, hematologic and genetic prognostic indicators; despite the consistent progress achieved in the understanding of the molecular pathogenesis and clinical treatment of AMLs, most patients ultimately fail with all these treatments, thus implying the absolute need to develop more efficacious treatment strategies.

The therapeutic potential of ascorbic acid (vitamin C) in leukemia has been known for many years. Particularly, the observation that leukemic patients display low vitamin C plasma levels due to increased uptake by the actively proliferating leukocytes [3] supports the rationale of using high doses of ascorbic acid not only as a prophylactic measure but also as a therapeutic approach for the treatment of leukemia and other cancers [4]. More recent studies have shown vitamin C deficiency in patients with myeloid malignancies and particularly in AML [5]. In fact, AML patients have shown a markedly lower plasma vitamin C concentration compared to normal controls; this was inversely correlated with the number of circulating blast cells, which were significantly decreased at the time of clinical remission and further decreased in relapsing/resistant disease [6]. The results obtained in AML patients are supported by data on the vitamin C concentration in leukemic cells, which is lower than in normal mononuclear cells [6]. However, no association has been observed between vitamin C levels and the mutational profile of AML patients [6]. Of interest, ascorbic acid is a key epigenetic regulator involved in hematopoietic stem cell function and leukemogenesis. Particularly, it has been shown that there is a clear relationship between the activity of TET enzymes, vitamin C and the development of hematopoietic malignancies [7,8]. The same reports have shown that vitamin C can protect hematopoietic stem cells (HSCs) from epigenetic alteration, which induces leukemic progression through the stimulation of the catalytic activity of TET2 [7,8]. Ascorbic acid is a double-edged bioactive molecule, proven to exert either an antioxidant effect at low doses (µM range) or a pro-oxidant effect at high pharmacological doses (mM range) achievable in vivo through intravenous administration (IV); at the highest doses, ascorbic acid induces apoptosis of leukemic cells [9]. Particularly, ascorbic acid at a pharmacologic concentration induces apoptosis of leukemic blasts, including primary leukemic blasts, with a preferential cytotoxic effect against leukemic cells, sparing normal hematopoietic cells, including hematopoietic progenitor and stem cells [10,11]. Therefore, leukemic cells seem to be more sensitive to oxidative stress than normal stem/progenitor cells.

The cancer-specific pro-oxidative cytotoxic effects, epigenetic regulation, immune modulation, inhibition of hypoxia and oncogenic kinase signaling support using vitamin C as a potential anti-leukemic drug or at least implementing it in current therapies due to its real multi-targeting effect [12,13,14]. However, despite the strong rationale, current clinical evidence supporting the therapeutic efficacy of high-dose intravenous (IV) vitamin C is ambiguous, and strong clinical data and phase III studies are lacking [12,13,14].

Some clinical studies have explored the clinical safety and efficacy of vitamin C in AML patients. Aldoss et al. reported the limited efficacy of ascorbic acid administered by IV route in association with arsenic trioxide in relapsing/refractory AML patients [15]. Other studies showed some efficacy of ascorbic acid in association with decitabine in the treatment of elderly AML patients [16]. In particular, *TET2*-mutated AMLs could represent a sensitive therapeutic target [17].

Plasma vitamin C concentrations vary substantially with the route of administration: peak plasma concentrations are markedly higher after administration of IV doses than after administration of oral doses [18]. Furthermore, the vitamin C half-life is short, and to maintain adequate in vivo concentrations, continuous infusions are required [19]. Finally, vitamin C pharmacokinetics is complex and influenced by many environmental and lifestyle factors [19]. Therefore, new vitamin C formulations would require improved bioavailability.

In this context, liposome-encapsulated vitamin C has achieved significantly higher and more stable levels than standard vitamin C after oral administration [20].

However, the bioavailability of synthetic vitamin C is far from satisfactory, thus generating problems when used at high dosages that too often lead to massive elimination of the administered vitamin C through the gut with numerous side effects [21,22]. All in all, the main scientific reason responsible for the unsatisfactory clinical results is the low availability of existing vitamin C-based compounds.

The discovery of extracellular vesicles (EVs) has provided new fuel for the understanding of how our bodies actually work; the organs of the human body are highly interconnected through EVs [23,24].

EVs naturally deliver molecules of various origin, including proteins, lipids and nucleic acids, but they are also able to deliver chemical and biological drugs throughout the body [24]. More recently, much attention has been paid to exosome-like plant-derived nanovesicles (ELPDNVs) [25], which have many similarities with human exosomes but at the same time important peculiarities. They deliver bioactive compounds that are crucial for metabolism (e.g., ascorbic acid, glutathione and SOD-1), which, when derived from organically grown fruits and vegetables, do not contain toxic or disease-related molecules. Moreover, while the production of human exosomes may have some problems with scalability due to being released by cultured cells, ELPDNVs may be obtained on a large scale [25]. ELPDNVs from various fruits have been shown to contain a series of antioxidants (e.g., ascorbic acid, catalase, glutathione and superoxide dismutase) and to have a measurable antioxidant capacity [25]. Recent studies have demonstrated that ascorbic acid, catalase, glutathione and superoxide dismutase isolated from citrus species exert in vitro anticancer activities [26].

In the present study, we explored the in vitro anti-leukemic properties of nanovesicles isolated from grapefruit. The results of this study support the potential therapeutic use of grapefruit-derived nanovesicles as potential anti-leukemic drugs, inducing oxidative stress on leukemic blasts.

## 2. Results

### 2.1. Characterization of Grapefruit-Derived Nanovesicles

The first set of experiments was aimed at verifying the structural properties of the EVs isolated from grapefruits. To this purpose, nanovesicles isolated from grapefruit juices underwent transmission electron microscopy (TEM) and nanoparticle tracking analysis (NTA). TEM analysis showed homogenous, typical, rounded, whole, and undamaged small EVs (Figure 1A). EVs’ sizes ranged from 50 to 100 nanometers (nm). A higher magnification showed the full integrity of the EVs, with the presence of an intact bilayer membrane visible as a thin white filament surrounding the electron-dense EVs (Figure 1B). At the same time, EVs underwent nanoparticle tracking analysis to determinate their distribution, number and size. As shown in Figure 1C, EVs isolated from grapefruit juice had the typical distribution of extracellular vesicles, with a mean size equal to 147 ± 11 nm. In the same EV population, the total antioxidant capacity (TAC) and enzymatic (catalase) and non-enzymatic antioxidants (ascorbic acid and glutathione) were measured in a sample of 10^12^ EVs. TAC allowed us to evaluate the cumulative and synergistic action of all antioxidants present in the analyzed nanovesicles.

This set of results shows that the EVs obtained from grapefruit juice were homogenously within the size range of nanovesicles. Thus, we decided to call them exosome-like plant-derived nanovesicles (ELPDNVs). Moreover, they contained measurable amounts of a series of antioxidants (ascorbic acid, catalase and glutathione), showing a potent antioxidant activity (Table 1).

### 2.2. Effects of Grapefruit-Derived Nanovesicles and Ascorbic Acid on Survival of Leukemic Cell Line

This part of the study was aimed at comparing the effect of grapefruit ELPDNVs and commercially available ascorbic acid on U937 and K562 leukemic cell lines. In a preliminary set of experiments, we comparatively evaluated the effect of various doses of ELPDNVs and of ascorbic acid on the growth of U937 cells and observed that ELPDNV doses corresponding to 1 × 10^8^ or 1 × 10^12^/mL exerted a significant inhibitory effect, while ascorbic acid (vitamin C) only markedly inhibited cell growth at the highest dose of 2 mM (Figure 2). It is interesting to point out that the content of ascorbic acid in 1 × 10^12^ ELPDNVs corresponded to 0.2 mM, suggesting that ELPDNVs increase the bioavailability of the ascorbic acid naturally contained within them.

The first result we obtained was on proliferation. Figure 3A shows that ELPDNVs inhibited the proliferation rate in both U937 (left panel) and K562 (right panel) cell lines treated with 1 × 10^12^ ELPDNVs but not 0.2 mM of vitamin C in a time-dependent manner, reaching the most significant inhibition after 96 h of incubation. This result was consistent with the cell viability of the cells at 96 h, where ELPDNVs induced a significant decrease in the percentage of viable cells, particularly in the K562 cell cultures, while vitamin C did not induce any significant decrease in viable cells (Figure 3B, right panel, *p* < 0.001).

This set of results shows that ELPDNVs exerted a clear anti-proliferative effect on both U937 and K562 leukemic cell lines that was always time-dependent, while commercially available vitamin C at 0.2 mM did not show any significant effect on the same cell lines.

### 2.3. Grapefruit-Derived Nanovesicle Treatment on Survival of Normal and Leukemic Blast Cells and Leukemic Cell Lines

A further set of experiments was aimed at verifying the effect of ELPDNVs on cell growth in either normal HPC (HPC) or AML blast cells cultures (AML). The main characteristics of the AML samples are shown in Table 2.

On the basis of the previous dose-response experiments, we used 1 × 10^12^ ELPDNVs that were added to the cell cultures every 2 days and analyzed after 24 h, 48 h and 72 h. The values were plotted as fold change compared to untreated cells (Ctrl). The results show that ELPDNVs did not affect the number of normal HPCs, while the number of AML blasts was significantly decreased at each time examined, reaching the lowest values at 72 h (Figure 4A) (significance: * *p* < 0.05; ** *p* < 0.01; and *** *p* < 0.001). In a separate set of experiments, we analyzed the effects of either 1 × 10^12^ nanovesicles or 0.2–2 mM of ascorbic acid against AML blast cells by cell counting at different time points (24 h, 48 h and 72 h) (Figure 4B). The results show that ELPDNVs induced a significant decrease in the AML blast cell counts at each time point, with the maximum effect at 72 h, while ascorbic acid showed an effect on the AML blasts only at the highest dosage (i.e., 2 mM).

These results show that ELPDNVs significantly reduced the number of AML blasts without affecting normal HPC and that ascorbic acid showed an effect only at the highest dose.

Thus, we tested the effect of ELPDNVs on different leukemic cell lines. To this purpose, we compared the proliferation rate of HL60, Hel and MV4-11 at 48 h and 96 h after treatment with 1 × 10^12^ nanovesicles. The results show that the proliferation of all the leukemic cell lines was significantly affected by ELPDNV treatment, with the lowest values observed at 96 h for all cell lines except MV4-11, which reached the most significant inhibition at 48 h (Figure 5).

### 2.4. Effect of Grapefruit-Derived Nanovesicles and Ascorbic Acid on ROS Production in Normal Cells, Leukemic Blast Cells and Leukemic Cell Lines

The aim of this set of experiments was to verify whether the effect of ELPDNVs against both leukemic blasts and leukemic cell lines was consistent with a redox imbalance. To this purpose, we analyzed the effect of either ELPDNVs or vitamin C on ROS production of normal HPC, AML blast cells and U937 cells. All these cells underwent labeling with DCFDA (20 mM) and treatment with 50 mM tert-butyl hydrogen peroxide (TBHP) with 1 × 10^12^ ELPDNVs or 0.2–2 mM of vitamin C. Cells were then analyzed after 15 min, 30 min and 3 h of incubation for the evaluation of ROS production. The results show that ELPDNVs increased the ROS levels in both AML blast cells and U937 cells but did not change ROS accumulation in normal cells. Vitamin C at the high concentration (2 mM), but not at the low concentration (0.2 mM), increased ROS levels in AML blasts, U937 cells and normal HPCs (Figure 6).

The concomitant evaluation of ROS production and cell proliferation clearly showed that in leukemic cells (leukemic blasts and leukemic cell lines), there is a direct relationship between the induction of ROS production and inhibition of cell proliferation. In fact, the low dose of vitamin C (0.2 mM) was unable to induce both stimulation of ROS production and inhibition of cell proliferation, while ELPDNVs as well as the high dose of ascorbic acid (2 mM) elicited increased ROS production associated with inhibition of cell proliferation (Figure 7). In normal HPCs, ELPDNVs did no induce any significant increase in ROS production and did not affect their proliferation (Figure 7). Interestingly, in AML blasts grown for 24 h in either the absence or presence of ELPDNVs and then analyzed for ROS production, there was evidence of increased ROS production in cells continuously exposed to exosomes.

## 3. Discussion

The current therapy for hematologic malignancies is effective with a high rate of 5-year survival but is associated with heavy side effects and an increased risk of development of a secondary malignancy. While new target therapies are opening a new path in this area, the clinical results are not entirely satisfactory and too often restricted to patients expressing targetable molecules [27,28].

In the last decade, research has focused on natural compounds as an alternative strategy for cancers in general and for hematologic malignancies in particular [29]. One example is ascorbic acid or ascorbate (ascorbic acid in solution), which has been used at high concentrations in the treatment of leukemias, both in vitro and in vivo [13]. However, the use of high-dosage ascorbate has shown some problems related to the pharmacokinetics of vitamin C, requiring frequent drug administration of an intravenous dose of at least 1 g/kg [30,31].

The use of ascorbic acid directly obtained from natural sources may offer some advantages with respect to the synthetic compound. Citrus fruits are the most recognized natural source of ascorbic acid, and recent research has shown that exosome-like nanovesicles obtained from various citrus fruits contain measurable levels of ascorbic acid [25]. Moreover, the ascorbic acid contained in citrus fruit-derived exosomes are complexed with other potent antioxidants within these nanovesicles [25], showing a potent anti-oxidant activity and thus supporting their potential effectiveness.

In this study, we tested the ability of grapefruit-derived exosomes to inhibit the growth of both human leukemic cells and leukemic patient-derived bone marrow blasts. We first analyzed and characterized the homogeneity of extracellular vesicles obtained from grapefruits, showing that the EVs obtained from grapefruit juice were homogenously distributed within the size range of nanovesicles and that they maintained their integrity when analyzed by both transmission electron microscopy and NTA. For this reason, we decided to call them exosome-like plant-derived nanovesicles (ELPDNVs). Moreover, ELPDNVs contain measurable amounts of several bioactive molecules, including vitamin C, showing a potent antioxidant activity.

Further experiments aimed at evaluating the effectiveness of ELPDNVs showed that ELPDNVs exerted a clear anti-proliferative effect in both U937 and K562 leukemic cell lines, and this effect was always time-dependent. Of interest, commercially available ascorbic acid induced anti-proliferative and cytotoxic effects against leukemic cell lines at the highest dosage (2 mM) only.

A further set of experiments was aimed at verifying the effect of ELPDNVs on cell growth in either normal HPC (HPC) or AML blast cells cultures (AML). Dose-response and time-dependent experiments were performed in which 1 × 10^12^ ELPDNVs were added to the cell cultures every 2 days at different time points. The results showed that ELPDNVs did not affect the number of normal cells, while the number of AML blasts was significantly decreased at each time period examined, reaching the lowest values at 72 h. In a separate set of experiments, we analyzed the effects of either ELPDNVs or commercially available ascorbic acid against AML blast cells by cell counting at different time points. We want to emphasize that this represented an ex vivo experiment performed in AML blasts from seven different AML patients matched for sex and characterized for both molecular and cytogenetic features (see Table 2). The results showed that ELPDNVs induced a significant decrease in the AML blast cell counts at each time point, with the maximum effect at 72 h, while ascorbic acid showed an effect on the AML blasts only with the highest dosage tested (2 mM). As a whole, our results showed that ELPDNVs significantly reduced the number of AML blasts without affecting normal HPC and that ascorbic acid showed an effect only at the highest dose. Thus, we tested the effectiveness of ELPDNVs on different leukemic cell lines. To this purpose, we compared the proliferation rate of HL60, Hel and MV4-11 at 48 h and 96 h after treatment with 1 × 10^12^ nanovesicles. The results showed that the proliferation of all leukemic cell lines was significantly affected by ELPDNV treatment. The lowest values were observed at 96 h for all cell lines except MV4-11, which reached the most significant inhibition at 48 h. With these results, we showed that ELPDNV treatment significantly reduced the proliferation rate of both AML blast cells and different leukemic cells lines. In order to understand the mechanism/s underlying these results, we set up experiments aimed at verifying whether the effect of ELPDNVs against both leukemic blasts and leukemic cell lines was consistent with a redox imbalance. To this purpose, we analyzed the early effect of either ELPDNVs or ascorbic acid on ROS production of normal HPC, AML blast cells and U937. The results showed that ELPDNVs increased the ROS levels in both AML blast cells and U937 cells, but they did not change ROS production in normal cells. These observations support the view that ELPDNVs, as well as ascorbic acid at high doses, induce an inhibitory effect on leukemic proliferation and promote leukemic cell death through a mechanism related to stimulation of ROS production. Through this action, ELPDNVs could promote a redox imbalance of leukemic cells through a mechanism conceivably related to the ascorbic acid contained in the ELPDNVs, thus also supporting leukemic cells’ peculiar vulnerability to oxidative stress [12,32]. ROS are important signaling molecules, and their production is increased in myeloid leukemic cells, thus rendering these cells exposed to oxidative stress [32]. Oxidative stress and metabolic rewiring toward oxidative metabolism represent two intimately intermingled features of leukemic cells and offer new therapeutic approaches potentially capable of eradicating leukemic stem cells [33]. Some anti-leukemic drugs exert their pharmacologic activity by altering the cellular redox homeostasis of leukemic cells, increasing ROS production, such as arsenic trioxide used in acute promyelocytic leukemia (APL) [34], cytarabine currently used in AML [35] and bortezomib used in different hematological malignancies [36]. ELPDNVs could be used in combination with these drugs to enhance anti-leukemia effects by eradicating chemo-resistant cells through a synergistic effect and promoting oxidative stress.

The wide spectrum of biologic activities of ascorbic acid supports two pharmacologic activities attractive for antileukemia therapy: a pro-oxidant cytotoxic effect at higher doses and epigenetic modulation at lower doses [37].

These results suggest that ELPDNVs could have a possible role in the development of ascorbic acid-based therapies for AMLs. In this context, a recent study by Mouchel et al. retrospectively evaluated a cohort of 431 AML patients regarding the effect of vitamin C and vitamin D supplementation in addition to a standard chemotherapy induction regimen; 262 patients received no vitamin supplementation, and 169 received vitamin supplementation. Vitamin C/D supplementation significantly improved overall survival of *NPM1*-mutant AML patients and reduced the incidence of bacterial or fungal infections [38]. According to the findings of this study, it is tempting to speculate that ELPDNV-based therapy could have a possible role as a supplementary therapy for *NPM1*-mutant AML patients undergoing chemotherapy induction treatment. 

## 4. Materials and Methods

### 4.1. Fruit Material

Grapefruits (*C. paradisi* Star Ruby) were purchased from a farm with an organic farming certification. The fruits were washed with water and sodium bicarbonate, peeled, and extracted with a fruit juice extractor. Fruit juices were stored at −80 °C.

### 4.2. Nanovesicles Isolation

Fruit juices were centrifuged at 500× *g* × 10 min; the supernatants were filtered with 100 µm filters and serially centrifugated at 2000× *g* for 20 min to eliminate cell debris and then at 15,000× *g* for 30 min to eliminate the fraction enriched in the microvesicles. The supernatants were subsequently ultracentrifuged in a Sorvall WX Ultracentrifuge Series (Thermo Fisher Scientific IT, via Ravenna 8, Rome, Italy) at 110,000× *g* for 1 h 30 min to collect the nanovesicles. The pellet was resuspended in an appropriate buffer for downstream analyses. A total of 1.00 ± 0.13 × 10^12^ exosomes were isolated from 11 mL of grapefruit extract.

### 4.3. Transmission Electron Microscopy (TEM)

After isolation, extracellular vesicles (EVs) resuspended in PBS and properly diluted were incubated for 5 min onto carbon-coated copper grids of 200 mesh (Electron Microscopy Sciences, Hatfield, PA, USA) at room temperature. Once absorbed on the grids, EVs were fixed with 2% glutaraldehyde in PBS (Electron Microscopy Sciences, Hatfield, PA, USA) for 10 min and then washed 3 times in Milli-Q water. Negative staining was performed with 2% phosphotungstic acid. Finally, the grids were air dried and observed using a CM 100 Philips (Philips Co., Amsterdam, The Netherland).

### 4.4. Nanoparticle Tracking Analysis

Nanoparticle tracking analysis (NTA) using the NanoSight NS300 (Malvern, Worcestershire, UK) was used for the measurement of size distribution and concentration of extracellular vesicle samples in the liquid suspension. Five videos of typically 60 s duration were taken. Data were analyzed using the NTA 3.0 software (Malvern Instruments), which was optimized to first identify and then track each particle on a frame-by-frame basis. The Brownian motion of each particle was tracked using the Stokes–Einstein equation: D° = kT/6πηr, where D° is the diffusion coefficient, kT/6πηr = f0 is the frictional coefficient of the particle for the special case of a spherical particle of radius r moving at a uniform velocity in a continuous fluid of viscosity η, k is Boltzmann’s constant, and T is the absolute temperature.

For EV analysis by NTA, the parameters were set as follows: camera level 15 and detection threshold 5.

### 4.5. Total Antioxidant Power Assay (PAO Test Kit)

Detection and quantification of total antioxidant capacity were performed in nanovesicles using a colorimetric assay with the PAO Test Kit for Total Antioxidant Capacity (JaICA, Shizuoka, Japan). The assay can detect not only hydrophilic antioxidants, such as vitamin C and glutathione, but also hydrophobic antioxidants, such as vitamin E. The determination of the antioxidant power was carried out using reduction of cupric ion (Cu^++^ to Cu^+^). Briefly, samples were incubated for 3 min at room temperature with a Cu^++^ solution, and the Cu^++^ was reduced by antioxidants to form Cu^+^, which reacts with a chromatic solution (bathocuproine) and was detected by absorbance at a wavelength of 480 to 490 nm. Antioxidant capacity was calculated from the Cu^+^ formed. Absorbance was recorded at 490 nm.

### 4.6. Ascorbic Acid Assay

Detection and quantification of ascorbic acid in nanovesicles were performed using a fluorometric Ascorbic Acid Assay Kit (Sigma-Aldrich, St. Louis, MO, USA). Samples were diluted in an ascorbic acid buffer in a 96-well plate, and subsequently, a catalyst was added to the reaction mix in each well (the reaction mix was composed of an ascorbic acid buffer, ascorbic acid probe and ascorbic acid enzyme mix). After 5 min of incubation, fluorescence was read using a microplate reader at Ex/Em = 535/587 nm.

### 4.7. Catalase Activity Assay

For the catalase activity assay, a fluorometric kit (Abcam, Cambridge, UK) was used for detection and quantification of the catalase activity in fruit-derived nanovesicles. Briefly, samples resuspended in PBS were loaded in a 96-well plate. A stop solution was added to the control samples, which were incubated for 5 min at 25 °C to inhibit the catalase activity. A catalase reaction mix (with H_2_O_2_) was added to both the control and high control samples for 30 min at 25 °C. The reaction in the high control samples and standard samples was stopped with the stop solution. The developer was added to all wells, and after 10 min, the fluorescence was read at Ex/Em = 535/587 nm on a microplate reader (Promega, Madison, WI, USA). Data were analyzed following the manufacturer’s instructions. One unit of catalase corresponded to the amount of catalase that would decompose 1 µmol of H_2_O_2_ per minute at pH 4.5 and 25 °C.

### 4.8. Reduced Glutathione (GSH) Detection and Quantification Assay

The Glutathione Colorimetric Detection Kit (Thermo Fisher, Waltham, MA, USA), a colorimetric assay, was used for detection and quantification of reduced glutathione (GSH) levels in plasma preparations. The detection reagent and reaction mixture (NADPH and glutathione reductase) were added to samples, and after 20 min of incubation at room temperature, the optical densities were recorded at 405 nm.

### 4.9. Human Cord Blood CD34+ HPC Purification and Culture In Vitro

Human cord blood (CB) was obtained from healthy, full-term placentas according to the institutional guidelines of A.Fa.R. Centro Trasfusionale, Università La Sapienza, Rome, Italy with approval by the local ethical committees of Istituto Superiore di Sanità, Rome (file number #171639). Low-density mononuclear cells were isolated, and CD34^+^ cells were purified by positive selection using the MACS immunomagnetic separation system (Miltenyi Biotec, Bergisch Gladabach, Germany) according to the manufacturer’s instructions. The purity of CD34^+^ cells assessed by flow cytometry was routinely over 95%. Each single experiment could include pooled cells derived from different (2/3) cords blood. CB CD34^+^ HPCs were maintained in culture in Iscove’s medium supplemented with 10% FCS, GM-CSF (10 ng/mL) (PeproTech Inc., Rocky Hill, NJ, USA), SCF (10 ng/mL) and IL-3 (10 ng/mL) (R&D Systems, Minneapolis, MN, USA) [39].

### 4.10. Human Primary AML Blasts Culture

Bone marrow (BM) samples were collected from 7 consecutive newly diagnosed de novo AML patients admitted to the Department of Hematology of the University of Rome Tor Vergata. All samples had at least a 70% infiltration by leukemic blasts. Written informed consent was obtained from all patients in accordance with the Declaration of Helsinki. 

This study was approved by the local ethical committees of Istituto Superiore di Sanità and University of Tor Vergata (file number RS 34.20 26 February 2020 Fondazione PTV Policlinico Tor Vergata, Rome). Freshly isolated human primary AML blasts were isolated using the same procedure used for purification of normal CB CD34^+^ and maintained in culture in Iscove’s medium supplemented with 10% FCS, GM-CSF (10 ng/mL) (PeproTech Inc., Rocky Hill, NJ, USA), SCF (10 ng/mL) and IL3 (10 ng/mL) (R&D Systems, Minneapolis, MN, USA), as described [31].

### 4.11. Genetic Characterization of Leukemic Samples

Conventional karyotyping was performed and reported according to the International System for Human Cytogenetic Nomenclature [40]. For molecular diagnostic studies, total RNA was extracted from bone marrow mononuclear cells separated by Ficoll–Hypaque using the method by Chomczynsky and Sacchi [41]. DNA were extracted using a column-based kit (QIAmp DNA, Quiagen, Hilden, Germany). Samples were characterized for the presence of BCR/ABL, PML/RARA, CBFB/MHY11, RUNX1/RUNX1T1 and DEK/CAN fusion genes and for NPM1, FLT3-internal tandem duplication (ITD), DNMT3A, cKIT, IDH1 and IDH2, as shown in Table 2.

### 4.12. Flow Cytometry Analysis

Analysis of cell surface antigens was performed by flow cytometry using an FACS Canto (Becton Dickinson, Bedford, MA, USA). The cells were stained with a PE-conjugated anti-mouse, CD34 antibody (Pharmingen, San Diego, CA, USA). Briefly, the cells were resuspended in a Ca^2+^ Mg^2+^-free phosphate buffered saline solution (PBS) containing 20% FCS and mouse IgG (40 μg/mL), which was incubated for 10 min on ice and then incubated for 30 min at 4 °C with an appropriate dilution of antibody. After three washings in PBS, the cells were fixed in PBS formaldehyde (4%) and analyzed by flow cytometry.

### 4.13. Cell Lines Samples

U937 (as a model of AML-M5), K562 (erythroleukemic cell line), HL-60 (as models of AML-M4), HEL (erythroleukemic cell line) and MV4-11 (as AML-M5 mutated for FLT3-ITD) were grown at 37 °C in a humidified atmosphere of 5% CO_2_ in air in an RPMI medium supplemented with 10% FCS (Gibco, Carlsbad, CA, USA).

### 4.14. Culture with Grapefruit-Derived Nanovesicles and Increasing Doses of Ascorbic Acid

Cell lines, AML blast cells and their normal counterparts (normal CD34^+^ cells) were exposed to fresh grapefruit exosome-like plant-derived nanovesicles (ELPDNVs) or 0.2 mM–2 mM of ascorbic acid (vitamin C) and cultured for 7 days as above reported. During the first days of culture, 1 × 10^12^ ELPDNVs were resuspended in an appropriate culture medium, or 0.2 mM–2 mM of vitamin C (dissolved in water) was added to 2.5 × 10^5^ cells every 2 days up to the conclusion of the experiment.

### 4.15. Cell Growth and Viability Analysis

Cell viability and proliferation were investigated in cell lines, normal HPC and leukemic blasts before and after treatments. Cell growth was analyzed by cell counting using trypan blue, and the rate of proliferation and the percentage of viable cells were analyzed.

### 4.16. Oxidative Stress Detection

To evaluate the oxidative stress, cells were treated with ELPDNVs or vitamin C, and then, the production of reactive oxygen species (ROS) in live cells was determined by an ROS assay kit (Abcam, Cambridge, UK) according to the manufacturer’s protocols.

### 4.17. Data Analysis and Statistics

Statistical significance was evaluated using GraphPad Prism^TM^ (San Diego, CA, USA) software version 6.0. All data reported were verified in at least 3 independent experiments and are reported as mean ± SEM. Test details of each experiment are described in the figure legends, and *p* values < 0.05 were considered significant.

## 5. Conclusions

The results of this study support the potential therapeutic use of grapefruit-derived nanovesicles as anti-leukemic drugs. ELPDNVs showed a clear anti-leukemic effect that was consistent with an increased redox imbalance within the treated cells, as shown by the increase in ROS levels. This kind of effect was typical of the double-edged behavior of ascorbic acid that exerts either an antioxidant effect at low doses (µM range) or a pro-oxidant effect at high pharmacological doses (mM range) achievable in vivo through intravenous (IV) administration of commercially available vitamin C. We knew that at the highest doses, ascorbic acid induces apoptosis of leukemic cells [9,10,11]. We provide clear evidence that the effects obtained with ELPDNVs were comparable with the effects obtained with the highest (2 mM) doses of commercially available vitamin C, thus supporting the central role of the vitamin C contained in the ELPDNVs we purified and concentrated from the organically grown grapefruits. These results have greater importance given the evidence that ELPDNVs by oral administration showed a clear in vivo effect in a model of H_2_O_2_-treated mice [42]. We want to emphasize the high level of bioavailability of all the bioactive compounds, including vitamin C, contained in the ELPDNVs [42] and also the importance of the fact that grapefruit-derived ELPDNVs come from organic farming. In fact, exosomes deliver beneficial molecules (including bioactive compounds and nucleic acids), and they belong to a scavenging framework aimed at eliminating unwanted material from living organisms. As evidence, the exosomes from organic farming are significantly more beneficial than those derived from intensive agriculture [24,25].

Thus, we definitively propose ELPDNVs from organically grown grapefruits as a combination/supporting therapy for human leukemias with the aim of improving the effectiveness and reducing the side effects of the existing chemotherapy protocols, particularly in those patients showing resistance to current therapies.

## Figures and Tables

**Figure 1 ijms-24-15663-f001:**
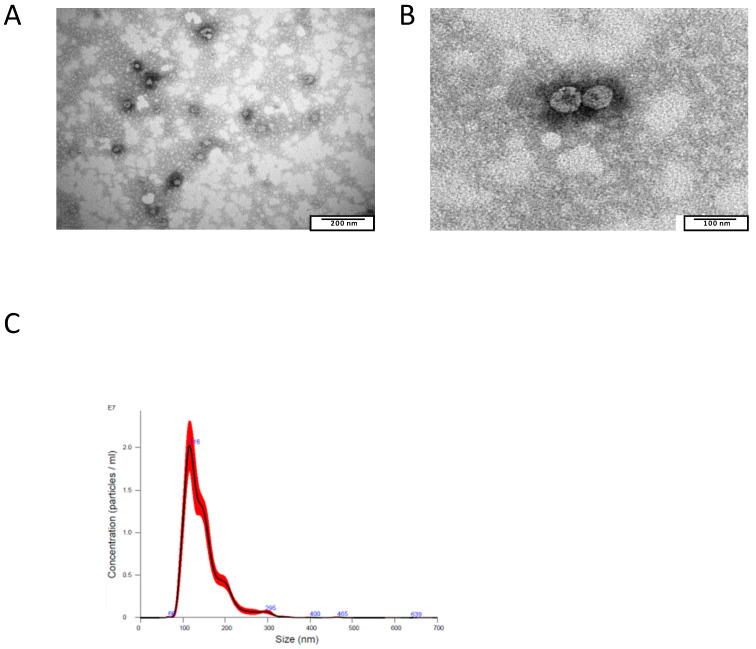
Characterization of grapefruit-derived nanovesicles. (**A**) Grapefruit nanovesicles were analyzed through transmission electron microscopy to evaluate their morphology and size. (**B**) Evaluation of nanovesicles’ membrane integrity through TEM analysis. (**C**) Isolated nanovesicles were analyzed through nanoparticle tracking analysis to evaluate their distribution profile and mean size.

**Figure 2 ijms-24-15663-f002:**
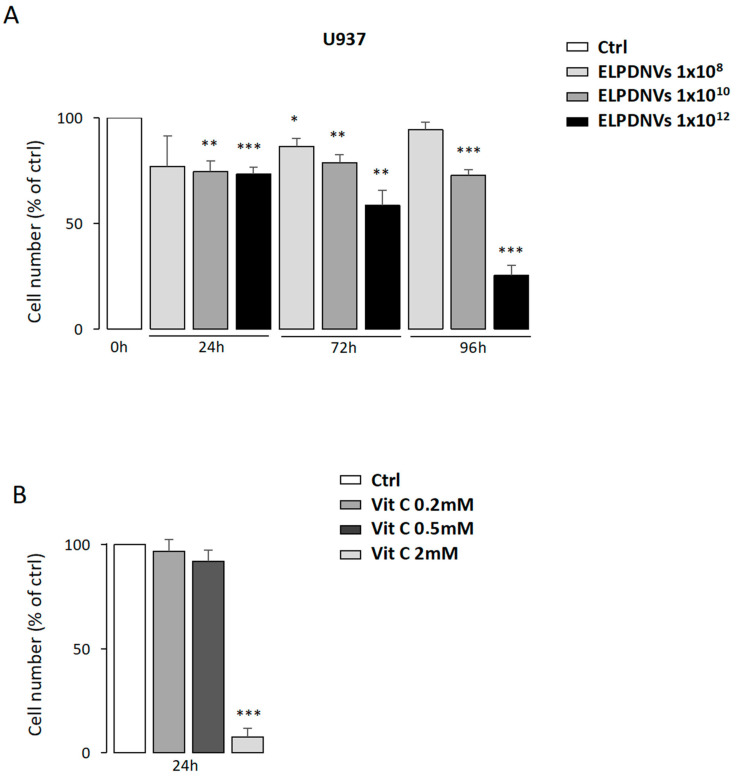
Effects of increasing doses of grapefruit-derived nanovesicles and ascorbic acid on cell growth. (**A**) Dose-response analysis of nanovesicle treatment on U937 cell growth compared to untreated cells (Ctrl). The results of three independent experiments (mean ± SEM values) are shown; Student’s *t*-test significance was * *p* < 0.05; ** *p* < 0.01; *** *p* < 0.001. (**B**) Reduced proliferation, induced by increasing concentrations of ascorbic acid (0.2–2 mM), was evaluated at 24 h in U937 cell lines. One representative experiment out of three is shown.

**Figure 3 ijms-24-15663-f003:**
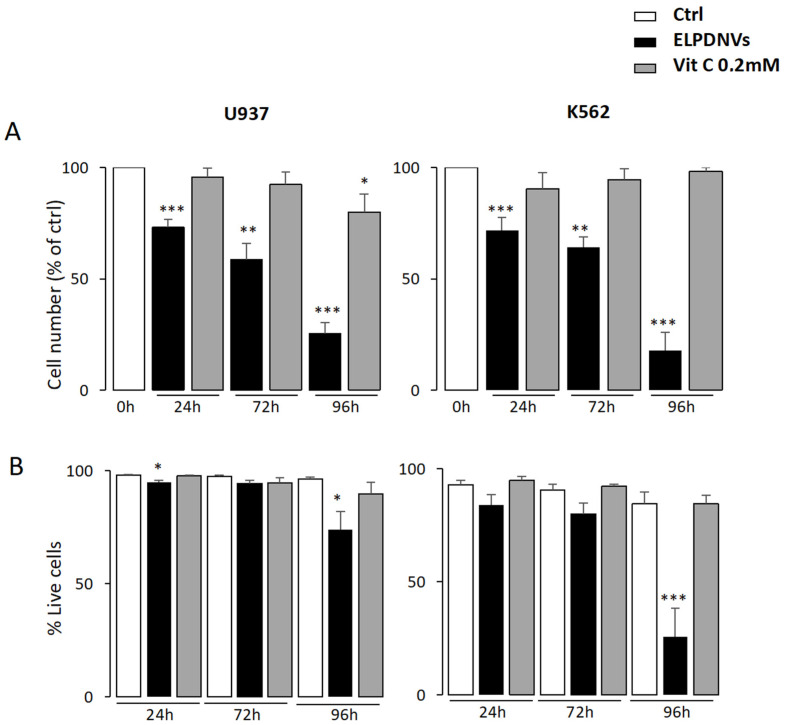
Effects of grapefruit-derived nanovesicles and ascorbic acid on the survival of leukemic cell lines. In U937 (**left panel**) and K562 (**right panel**) cells treated with 1 × 10^12^ nanovesicles or 0.2 mM of ascorbic acid, the rate of proliferation (**A**) and the percentage of viable cells (**B**) were analyzed by cell counting after 24 h, 72 h and 96 h. The values were plotted as fold change compared to untreated cells (Ctrl). Values are expressed as mean ± SEM of three independent experiments. Student’s *t*-test significance was * *p* < 0.05; ** *p* < 0.01; *** *p* < 0.001.

**Figure 4 ijms-24-15663-f004:**
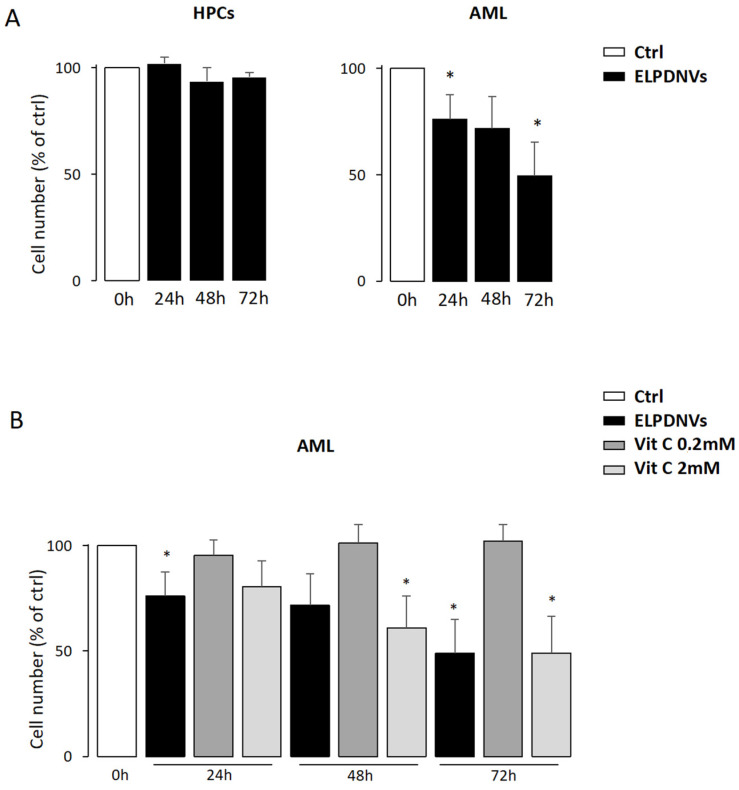
Grapefruit-derived nanovesicle treatment on survival of normal and leukemic blast cells. (**A**) Cell growth analysis of normal HPC (HPC) or AML blast cells (AML) in the presence of 1 × 10^12^ nanovesicles added every 2 days in cultures and analyzed after 24 h, 48 h and 72 h. The values were plotted as fold change compared to untreated cells (Ctrl). (**B**) AML blast cells treated with 1 × 10^12^ nanovesicles or 0.2–2 mM of ascorbic acid and analyzed by cell counting after 24 h, 48 h and 72 h. The values were plotted as fold change compared to untreated cells (Ctrl). Values are expressed as mean ± SEM of three independent experiments. Student’s *t*-test significance was * *p* < 0.05.

**Figure 5 ijms-24-15663-f005:**
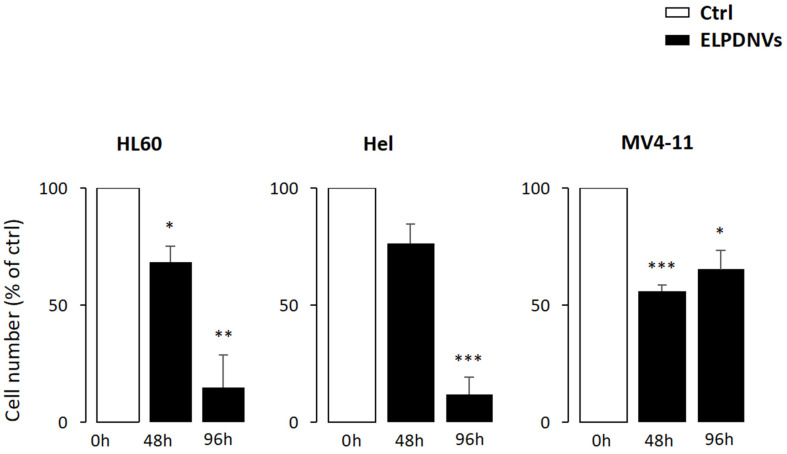
Grapefruit-derived nanovesicles inhibit the growth of leukemic cell lines. In HL60, Hel and MV4-11, the rate of cell growth was measured after 48 h and 96 h of treatment with 1 × 10^12^ ELPDNVs. The values were plotted as fold change compared to untreated cells (Ctrl) and were expressed as mean ± SEM of three independent experiments. Student’s *t*-test significance was * *p* < 0.05; ** *p* < 0.01; *** *p* < 0.001.

**Figure 6 ijms-24-15663-f006:**
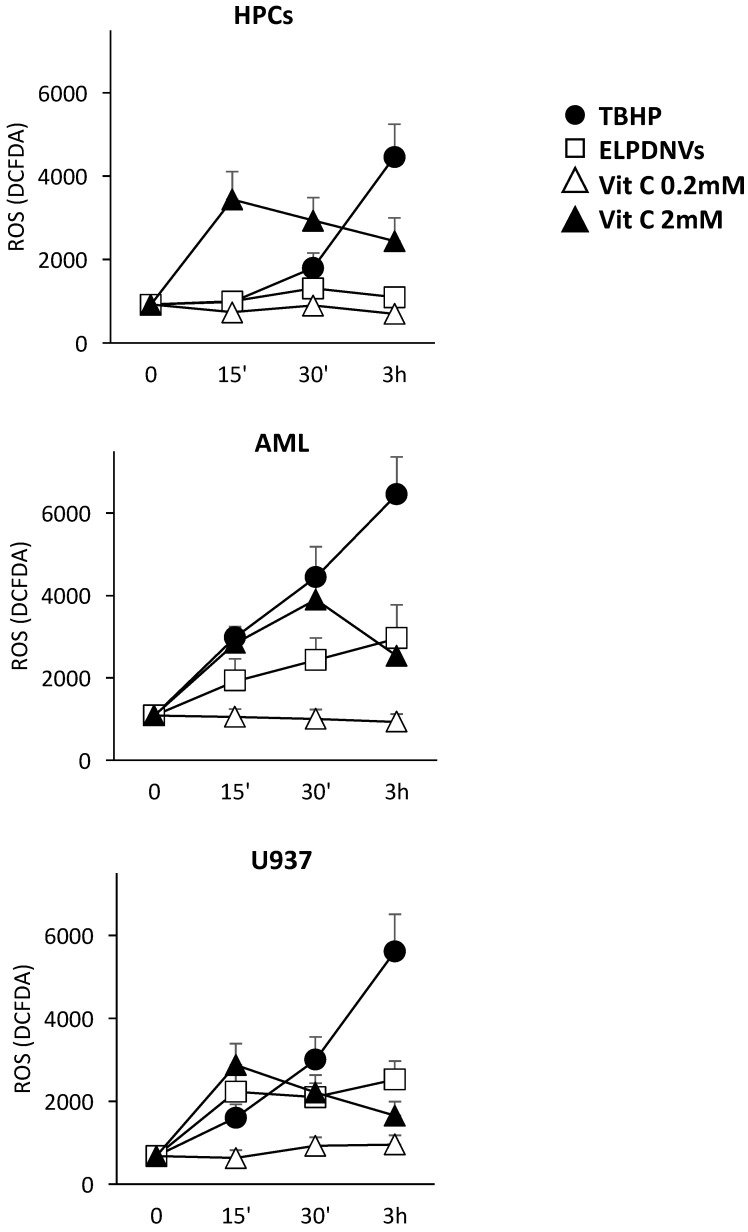
Effect of grapefruit-derived nanovesicles and ascorbic acid on ROS production in normal cells, leukemic blast cells and leukemic cell line U937. Normal HPCs, AML blast cells and U937 cells were labeled with DCFDA (20 mM) and then treated with 50 mM tert-butyl hydrogen peroxide (TBHP) as a positive control and 1 × 10^12^ ELPDNVs or vitamin C according to the protocol. Cells were then analyzed after 15 min, 30 min and 3 h of incubation for the evaluation of ROS production. Mean value ± SEM of three different experiments for HPCs and U937 cells. Mean value ± SEM observed for fresh AML blasts of five different patients.

**Figure 7 ijms-24-15663-f007:**
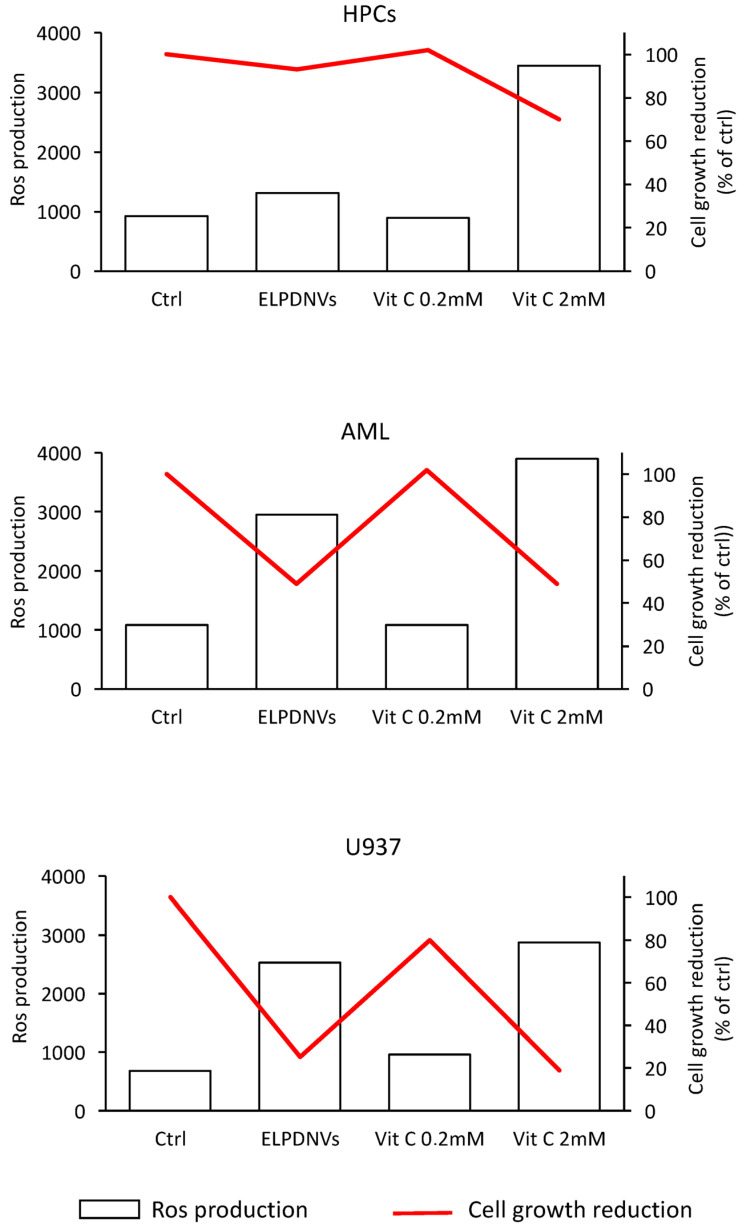
ROS production and cell growth in normal HPCs, AML blasts and U937 cells. For each cell type, the highest value of ROS production and the highest reduction in cell growth is reported compared to the respective basal control cells. ROS production was measured as reported in Figure 6.

**Table 1 ijms-24-15663-t001:** Antioxidant content of grapefruit-derived nanovesicles.

Characterization of Antioxidant Content	
**Total antioxidant capacity**	12.5 ± 0.4 mM
**Ascorbic acid**	170 ± 4.1 ng
**Catalase**	12.4 ± 4.3 U/mL
**Glutathione**	1.2 ± 0.14 mM

**Table 2 ijms-24-15663-t002:** Characteristics and main molecular and cytogenetic abnormalities of AML samples.

Patient	Sex	% Blasts	Molecular Biology	Cytogenetic
1	M	95	RUNX1/RUNX1T1; cKIT	46,XY,t(8;21)(q22;q22)
2	M	30	Negative panel	ND
3	M	20	Negative panel	46,XY
4	F	12	NPM1	ND
5	F	99	FLT3-ITD	46,XX,del(7)(q22) [15]
6	F	98	FLT3-ITD	46,XX
7	M	92	Negative panel	47,XY,trisomy(10),del(q20)

ND: not done.

## Data Availability

Not applicable.

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
