# Peer review of "Ex Vivo Anti-Leukemic Effect of Exosome-like Grapefruit-Derived Nanovesicles from Organic Farming—The Potential Role of Ascorbic Acid"

_ijms, 2023, doi:10.3390/ijms242115663_

Round 1
Reviewer 1 Report
Comments and Suggestions for Authors
The manuscript investigates the effect of extracellular vesicles derived from grapefruit on the cell growth, proliferation and oxidative stress in different models of leukemia. In general, the manuscript requires substantial improvement in terms of data presentation. The following aspects need to be particularly addressed:
1. Nanovesicle isolation. Please, specify the amount of starting material (volume of fruit juice).
2. Section 4.4 (NTA). Please, indicate the camera level and threshold used for EV analysis.
3. Line 440. It is not clear what the authors mean with “normal counterpart”. Please, clarify.
4. Section 4.16. Please, specify the principle of the kit used for ROS detection.
5. Table 1. Units (concentration/total amount) should be consistent between the different parameters. In addition, data should be better normalized to the number of vesicles.
6. Supplementary Figure 1B. Data needs to be presented as the mean of the three experiments performed and a statistical analysis has to be performed.
7. If Supplementary Figure 1B will appear in the manuscript, as in the current version, it should be renamed to figure 2.
8. Figure 2. It is not clear how “cell proliferation” was measured in the panel A. According to the y-axis, “Cell number” is shown. Please, clarify.
9. Table 2. Please, define the abbreviations used in the table.
10. Line 234. It is not clear what the authors mean with “almost three independent experiments”. The “n” of the different experimental groups should be specified in the figure legend.
11. Figure 4. Here, again, it is not clear, how the rate of cell proliferation was calculated. According to the y-axis, the “cell number” is depicted.
12. Figure 5. Data corresponding to the three experiments need to be shown and statistical analysis needs to be performed in order to support the conclusions.
13. Figure 6. Data presented in this figure is not clear at all. The concept of “peak of proliferation reduction” is difficult to understand. It is not clear, which parameter is being depicted. In addition, what do the authors mean with “peak value of ROS”. All parameters need to be presented as the mean +/- SEM or SD for all the experiments performed. Furthermore, to support the conclusions, a statistical analysis needs to be performed.
Comments on the Quality of English Language
Moderate English editing required
Author Response
Reply to reviewer 1
- The amount of starting material, as volume of fruit juice, was now specified. Specifically, 1.00 ±13 x 1012 exosomes were isolated from 11 ml of grapefruit extract.
- The camera level and threshold used for EV analysis were now shown. Particularly: For EV analysis by NTA the parameters were set as follows: camera level: 15 and detection threshold: 5.
- Line 440, it is now shown that normal counterpart indicates normal CD34+
- Section 4.16, it is now indicated the principle of the kit used for ROS detection.
- In Table 1, now units are unified.
- The data included in Fig. 1B are now reported as the mean value of three different experiments and the results of statistical evaluation are indicated.
- 1B was now renamed as Fig.2.
- In panel A of Fig.2, now Fig.3 in the revised manuscript, cell proliferation was measured as cell growth, counting the number of viable cells at each day of culture.
- Abbreviations used in Table 2 are now defined (see Table legend).
- Line 234, by a mistake it was reported almost in three experiments, but in reality, it was at least in three experiments. The n of the different experimental groups was now reported in Figure 4 legend.
- 4, now Fig.5 in the revised manuscript. As in Fig.2, cell proliferation indicates an analysis of cell growth. Thus, the term cell proliferation was replaced by cell growth.
- Data corresponding to three experiments are now shown in Fig. 5.
- We have tried to clarify data presented in Fig.6 (now Fig.7 in the revised manuscript). Thus, it is now indicated that the Figure reports the highest values of ROS production and the highest reduction of cell growth for normal HPC cells, leukemic AML blasts and U937 cells, respectively.
Reviewer 2 Report
Comments and Suggestions for Authors
Castelli et al. have discovered a potential of natural produce to kill leukaemia cells. The concept is interesting and follows logical path of overcoming limitations of conventional Vitamin C treatment for cancer, and connecting it to oxidative stress.
However, there are major concerns with experimental design and some crucial method description.
1. The juice and supernatant derived from grapefruit juice should be used as an experimental control for ELPDNV´s anti-cancer effect.
2. AML patient sample information is completely missing. Are the AML blasts isolated from one or many (n=
?) AML patient samples: Were the leukaemic blasts frozen or freshly isolated?
3. Was there a reason to select cell lines including K562, U937, HL60, MV4-11? They all have distinct genetic characteristics and ELPDNV´s effect could have been correlated with the genetic features.
Minor concern
1. Use full name of NTA method in abstract
2. Speculate in conclusion which current therapy would benefit from combining ELPDNVs
3. Refer oxidative stress research articles in AML and comment in light of your findings in conclusion
Author Response
Reply to reviewer 2
- We understand the point, but the juice per se cannot be used in culture conditions in that it dramatically reduces the pH of the culture medium and it adds fruits’ sugar that can fuel the tumor cells.
- AML patient information was now detailed as requested, in the section 4.10 of the manuscript.
- Cell lines K562, U937, HL60 and MV4-11 have been selected because they correspond to different cell differentiation pathways (K562 erythroleukemic, U937 promonocytic, HL60 myeloblastic, MV4-11 myelomonocytic), are currently used in experimental studies and are well characterized at immunophenotypically and molecular level.
- Full name of NTA (nanoparticle tracking analysis) method in the abstract was now defined.
- At the end of discussion, a speculation was added about a possible benefit that could derive from combining ELPDNVs with current therapy, particularly in NPM1-mutant AML patients.
- Oxidative stress research articles in AML have been refered and analyzed in the context of our findings (see the end of discussion).
Round 2
Reviewer 1 Report
Comments and Suggestions for Authors
The authors have addressed most of my concerns and delivered an improved version. A few aspects still need to be modified/improved before publication:
1. Table 1. The units of the acid ascorbic are still unclear. While all the other parameters are reported as concentration, ascorbic acid is reported as total mass. Without the information of the volume, it is very difficult to understand this information.
2. Please, add information in the section 4.17 or in each figure legend about the statistical tests performed.
3. Line 222 of the revised version. The reference to Fig 2B is wrong (should be 3B).
4. Line 439. There are some typos in “vitamin C7D supplementatio”
Comments on the Quality of English LanguageMinor editing of English language required
Author Response
Reply to Referee 1
- At the end of section 4.2 it was reported that the volume of grapefruit extracts is 11mL.
- Information was now added in each figure legend about the statistical test used for comparison of different experimental groups.
- Line 222, the reference erroneously indicated as Fig.2B was changed with Fig. 3B.
- Line 439, typos were corrected.
Reviewer 2 Report
Comments and Suggestions for Authors
All the comments are answered satisfactorily.
Author Response
OK